# Triple-Negative Breast Cancer Intrinsic FTSJ1 Favors Tumor Progression and Attenuates CD8+ T Cell Infiltration

**DOI:** 10.3390/cancers16030597

**Published:** 2024-01-31

**Authors:** Yangqing Sun, Qingqing Liu, Shangwei Zhong, Rui Wei, Jun-Li Luo

**Affiliations:** 1Department of Oncology, Xiangya Hospital, Central South University, Changsha 410008, China; 198101050@csu.edu.cn (Y.S.); liuqq_2018@csu.edu.cn (Q.L.); 2The Cancer Research Institute and the Second Affiliated Hospital, Hengyang Medical School, University of South China, Hengyang 421001, China; swzhong@usc.edu.cn; 3National Health Commission Key Laboratory of Birth Defect Research and Prevention, Hunan Provincial Maternal and Child Health Care Hospital, Changsha 410008, China

**Keywords:** FTSJ1, tumor promotor, CD8+T cell infiltration, triple-negative breast cancer

## Abstract

**Simple Summary:**

In this study, we found that high FTSJ1 expression in triple-negative breast cancer patients was associated with poor prognosis and was associated with reduced infiltration of CD8+T cells in the tumor microenvironment. By knocking down FTSJ1, we observed an inhibitory effect on the proliferation and migration of triple-negative breast cancer, while inducing apoptosis and increasing the sensitivity of TNBC cells to T-cell-mediated cytotoxicity. This finding highlights the importance of FTSJ1 as a potential immunotherapy target in triple-negative breast cancer.

**Abstract:**

FtsJ RNA 2′-O-methyltransferase 1 (FTSJ1) is a member of the methyltransferase superfamily and is involved in the processing and modification of ribosomal RNA. We herein demonstrate that FTSJ1 favors TNBC progression. The knockdown of FTSJ1 inhibits TNBC cell proliferation and development, induces apoptosis of cancer cells, and increases the sensitivity of TNBC cells to T-cell-mediated cytotoxicity. Furthermore, the high expression of FTSJ1 in TNBC attenuates CD8+T cell infiltration in the tumor microenvironment (TME) correlated with poorer prognosis for clinical TNBC patients. In this study, we establish that FTSJ1 acts as a tumor promotor, is involved in cancer immune evasion, and may serve as a potential immunotherapy target in TNBC.

## 1. Introduction

Up to 2020, the global cancer case data released by the International Agency for Research on Cancer (IARC) has attracted wide attention. Among all types of cancer, the number of new cases of breast cancer (BRCA) exceeded lung cancer for the first time, reaching 2.26 million cases, becoming the cancer with the highest incidence worldwide, so it was called the “world’s first cancer” title. This trend indicates that breast cancer has the highest incidence and mortality among women worldwide [1].

Triple-negative breast cancer (TNBC) commonly occurs in young women and accounts for about 10–15% of all breast cancers, shows strong invasiveness, high recurrence tendency, and poor prognosis compared with other types of breast cancer. TNBC is a subtype of breast cancer characterized by a lack of expression of three key receptors: estrogen receptor (ER), progesterone receptor (PR), and human epidermal growth factor receptor 2 (HER2). Lacking these receptors, TNBC has a limited response to hormonal therapy (because of the lack of ER and PR) and HER2-targeted therapies. Consequently, chemotherapy remains the main systemic treatment option for TNBC, while studies aimed at developing effective targeted therapies for this subtype and related to immunotherapy are ongoing [2,3,4].

The interactions occurring between TNBC cells and the nearby TME are crucial factors influencing the onset, advancement, and spread of TNBC [5,6,7]. Elevated levels of tumor-infiltrating lymphocytes (TILs) were indicative of positive outcomes in response to preoperative chemotherapy across various BRCA molecular subtypes studied. Additionally, higher TIL concentrations correlated with improved survival rates among TNBC [5]. Hence, targeting the mechanisms responsible for tumor-derived factors promoting infiltration and activation of lymphocytes could be viewed as a promising therapeutic approach for treating TNBC.

Studies have shown that modifications of tRNA bases play an important role in driving cancer phenotype. The tRNA methyltransferase, for example, can provide an additional level of gene regulation and has been shown to modulate the expression of specific proteins. The deletion of yeast tRNA methyltransferase 9 homolog (Trm9) leads to a decrease in the translation level of AGA and GAA codon-rich transcripts corresponding to the large ribonucleotide reductase subunit, which attenuates the efficiency of the responses to DNA damage and promotes tumor growth [8]. The tRNA methyltransferase 9-like protein (hTRM9L/KIAA1456) inhibits tumor growth via Lin-9 DREAM MuvB core complex component (LIN9) and hypoxia-inducible factor 1 subunit alpha (HIF1α)-dependent mechanisms [9]. Elongator complex proteins 1 and 3 (Elp1 and Elp3) and their partners in U34 tRNA modification, the cytosolic thiouridylases subunits 1 and 2 (Ctu1 and Ctu2), were found to influence the codon preference of transcripts decoding the proteins involved in melanoma survival [10]. The genetic ablation of Elp3 dramatically reduces breast cancer metastases in PyMT-induced breast cancer mouse models and affects cancer cell invasion in a mammary tumor ex vivo culture system [11].

FTSJ1 (FtsJ RNA 2′-O-methyltransferase 1) is a member of the methyltransferase superfamily. It produces proteins situated in the nucleolus and plays a role in the modification and processing of ribosomal RNA. Research has revealed the generation of diverse transcription variants through alternative splicing. Conditions linked to this gene are connected to cognitive impairment [12,13,14,15]. The impact of mutations or abnormalities in FTSJ1 on the efficiency of codon-specific translation and the plasticity of brain neurons has been confirmed through genetically modified mouse models [13,15].

The role of FTSJ1 in tumorigenesis and development is poorly studied and remains unclear. Studies have shown that FTSJ1 inhibits the malignancy of non-small cell lung cancer (NSCLC) by regulating tRNA 2′-O-methyladenosine modification and inhibiting the expression of DNA-damage-regulated autophagy regulator 1 (DRAM1) [16]. In liver cancer, however, FTSJ1 is one of the targets of the p53-mediated inhibition [17].

Overall, FTSJ1 has been poorly and ambivalently studied in cancer, and the role of FTSJ1 in the development of breast cancer and related TME is unknown. Here, we explore the role of FTSJ1 in the progression of TNBC cells, TME, and clinical outcomes and treatment of TNBC. Our findings suggest that FTSJ1 might serve as a potential therapeutic target in TNBC.

## 2. Materials and Methods

### 2.1. Public Database

Using UALCAN (http://ualcan.path.uab.edu/, accessed on 13 December 2023) provides the TCGA data, contrast analysis of FTSJ1 genes in cancer tissue and normal tissue, as well as in different subtype BRCA transcript expression differences [18,19].

The RNA sequencing expression profile of FTSJ1 and its corresponding clinical information were downloaded from the TCGA database (https://portal.gdc.com, accessed on 13 December 2023). The FTSJ1-low-score group was compared to the FTSJ1-high-score group using gene set variation analysis (GSVA) to evaluate the scores of hallmark gene sets from the MSigDB database [20,21,22].

The TCGA database was used to create precise immunological features of tumor-infiltrating cells based on ssGSEA algorithm.

Overall survival (OS) Kaplan–Meier (KM) plots of FTSJ1 in BRCA were generated using the GEPIA2.0 “Survival Analysis” module (http://gepia2.cancer-pku.cn/, accessed on 23 November 2023) [23].

### 2.2. Cell Culture

MDA-MB-231/GFP, HCC1937, and 4T1 cell lines were acquired from Procell (Wuhan, China). Ten percent of fetal bovine serum (FBS) (Pricella, 164210-50, Procell Life Science & Technology Co., Ltd., Wuhan, China) DMEM medium (Pricella, PM150210) was used to culture the MDA-MB-231/GFP (Pricella) and ten percent of FBS (Pricella, 164210-50) RPMI-1640 medium (Pricella, PM150110) was used to culture the HCC1937 (Pricella) and 4T1 (Pricella) at 37 °C and 5% CO_2_.

### 2.3. Plasmid and Lentivirus Preparation

The forward primer, GG (protective base) GGTACC (restriction site) CC (protective base) GCCACC (Kozak sequence) ATGTATCCTTACGACGTGCCTGACTACGCCATGGGACGGACGTCAAAG (partial HA tag), and the reverse primer ATAGTTTA (protective base) GCGGCCGC (restriction site) ATTCTTAT (protective base) TTAAGGTGAACAACTCATTTCA (partial sequence at the C-terminal of FTSJ1) were used to amplify the FTSJ1 from cDNA of MDA-MB-231 cell. The PCR product was then inserted into LVCV-19 vector (Sino Biological, Inc., Beijing, China) using the KpnI/NotI (New England BioLabs, Beijing, China). ShRNA coding sequences directed against human FTSJ1 mRNA were designed (the forward primer, 5′-CCGGCCATTCTTACGACCCAGATTTCTCGAGAAATCTGGGTCGTAAGAATGGTTTTTG-3′, and the reverse primer 5′-AATTCAAAAACCATTCTTACGACCCAGATTTCTCGAGAAATCTGGGTCGTAAGAATGG-3′) and inserted into lentiviral vector plasmid pLKO.1. ShRNA coding sequences directed against mouse FTSJ1 mRNA were designed (the forward primer, 5′-CCGGCCAACTCTTCAAAGGTGTGAACTCGAGTTCACACCTTTGAAGAGTTGGTTTTTG-3′, and the reverse primer 5′-AATTCAAAAACCAACTCTTCAAAGGTGTGAACTCGAGTTCACACCTTTGAAGAGTTGG-3′) and inserted into lentiviral vector plasmid pLKO.1. The FTSJ1 overexpression (FTSJ1 KD) plasmid (LVCV-19-FTSJ1), overexpression control (OE-control) plasmid (LVCV-19 vector), FTSJ1 knockdown (FTSJ1 KD) plasmid (pLKO.1-FTSJ1), and knockdown control (KD-control) plasmid (pLKO.1 vector) plus packaging plasmids (pMD2G and psPAX2) were transfected into 293T cells. Then, 48 h later, the supernatant was collected to harvest the lentiviral particles.

### 2.4. Establishment of Stable TNBC Cell Lines

TNBC cells were transduced with lentiviruses carrying FTSJ1 KD, KD control, FTSJ1 OE, and OE control. Following this, the cells were seeded in 10 cm cell culture dishes and grown in complete medium supplemented with puromycin (Sangon Biotech, Shanghai, China) 48 h post-transduction. Isolated single colonies were expanded in 96-well plates.

### 2.5. Cell Proliferation Analysis

PLKO plasmid (Addgene) encoding shRNA against FTSJ1 (sh-FTSJ1), or control shRNA was used for cell transfection with polyethylenimine (PEI) (Sigma, Osterode am Harz, Germany). After 6 h, the culture medium was renewed and cells were further cultured for 48 h, followed by being incubated in 96-well plates (10^3^ cells per well). Then, 10 µL of the Cell Counting Kit-8 (CCK8) (APExBIO, K1018, Houston, TX, USA) solution was added to each well for 2 h at a fixed time of 1, 3, 5 days after inoculation. The absorbance value of each well at 450 nm was measured for plotting a growth curve. Each experiment was repeated independently in triplicate.

### 2.6. Wound Healing Assay

Cells cultured in 12-well plates at 100% confluence were scratched using a 20 µL pipette tip. After completing the scratching, the cells underwent a gentle washing process three times with sterile PBS to ensure that the gap between scratches was clearly visible to the naked eye. The scratch time was defined as 0 h and cell images were recorded with a microscope. After recording, the cells were cultured in medium without FBS for another 24 h, and then the cells’ healing was again recorded by microscope. The image was opened using ImageJ 1.50i (Bethesda, MD, USA), and the scratch distance was calculated. The cell wound healing rate was determined using the formula:Cell wound healing rate (%) = (1 − wound distance at the indicated time point/wound distance at 0 h) × 100%.

### 2.7. Real-Time PCR

For RT-qPCR, a total of 1 μg of RNA was extracted using TRIzol reagents (Thermo Fisher Scientific (Waltham, MA USA), Cat#15596026) and employed for the reverse transcription reaction, utilizing a TAKARA reverse transcription kit (Takara, Beijing, China), following the manufacturer’s instructions. Following that, a PCR master mix from Vazyme (Nanjing, China) was used, and the detection was conducted using Bio-Rad CFX Maestro (Hercules, CA, USA). The primer sequences employed were as follows:CD276-F: 5′-CTGGCTTTCGTGTGCTGGAGAA-3′;CD276-R:5′-GCTGTCAGAGTGTTTCAGAGGC-3′;CD252 (TNFSF4)-F: 5′-CCTACATCTGCCTGCACTTCTC-3′;CD252 (TNFSF4)-R: 5′-TGATGACTGAGTTGTTCTGCACC-3′;TNFSF9-F: 5′-GGCTGGAGTCTACTATGTCTTCT-3′;TNFSF9-R: 5′-CGTGTCCTCTTTGTAGCTCAGG-3′;CD70-F: 5′-GCTTTGGTCCCATTGGTCG-3′;CD70-R: 5′-CGTCCCACCCAAGTGACTC-3′;CD274 (PD-L1)-F: 5′-TGGCATTTGCTGAACGCATTT-3′;CD274 (PD-L1)-R: 5′-TGCAGCCAGGTCTAATTGTTTT-3′;GAPDH-F: 5′-CTGGGCTACACTGAGCACC-3′;GAPDH-R: 5′-AAGTGGTCGTTGAG GGCAATG-3′.

The comparison of relative mRNA expression of genes was calculated by 2^−ΔΔCt^ method (Ct represents cycle threshold). Calculation method:ΔΔCt = (Ct (gene) − Ct (GAPDH)) under test sample − (Ct (gene) − Ct (GAPDH)).

### 2.8. Western Blot Analysis

Cellular proteins were extracted from TNBC cells using RIPA lysis buffer containing 1 mM PMSF and a cocktail. Following extraction, the proteins underwent electrophoresis and were subsequently transferred to PVDF membranes. Then TBST-dissolved 5% skim milk powder was blocked for 2 h at room temperature; after blocking, the membranes were exposed to the relevant antibodies and incubated overnight at 4 °C, FTSJ1 (1:1000, abcam (Waltham, MA, USA), ab227259), β-tubulin (1:1500, Huabio (Beijing, China), M1305-2), GAPDH (1:1000, Cell Signaling Technology (Danvers, MA, USA), #5174). These membranes were then incubated with the corresponding secondary antibodies for 2 h at room temperature. An appropriate amount of ECL luminescence working solution (Adansta, San Jose, CA, USA) was uniformly added. Image Lab 6.0.1 software (Bio-Rad, Hercules, CA, USA) was used for scanning development, and the strip scan map was saved. The grayscale values of the blots were quantified using ImageJ 1.50i (USA).

### 2.9. T Cell Separation, Activation, and Culture

Briefly, blood samples of healthy donors were used to isolate peripheral blood mononuclear cells (PBMCs). Fresh blood samples were diluted with phosphate-buffered saline (PBS) and separated using human lymphocyte separation medium (Cat: LTS1077, TBDsciences, Tianjin, China) via gradient centrifugation. T cells pre-stimulated with 500 ng/mL anti-human CD28 mAb (Cat: GMP-TL102, T&L Biological Technology, Beijing, China,) and 500 ng/mL anti-human CD3 mAb (Cat: GMP-TL101, T&L Biological Technology, Beijing, China), RPMI-1640 medium (Pricella) supplemented with 10 ng/mL recombinant human IL-2 protein (Cat: GMP-TL906, T&L Biological Technology, Beijing, China) was used for human T cell culture.

### 2.10. T Cell Cytotoxicity Assays

T cells were isolated and purified from PBMCs of the same donor. A total of 5000 MDA-MB-231-GFP cells were seeded into 1 well of 96-well plates and 8–10 replicates per group (FTSJ1 KD, control KD, FTSJ1 OE, control OE). Activated T cells were added to the upper layer of tumor cells in varying “effector to target” (E/T) ratios (0:1, 10:1, 20:1). After 24–48 h of co-culture, fluorescent cells were observed by fluorescence microscope, photographed, counted, and analyzed.

### 2.11. Xenograft Tumor Mouse Models

A total of 5 × 10^5^ 4T1 cell lines were subcutaneously grafted into 6-week-old female NOD-SCID or BALB/c mice (from SJA Laboratory Animal Ltd., Changsha, China). The long diameter (length) and short diameter (width) of the tumor were measured twice a week starting after the 5th day of injection. The tumor volume was calculated using the formula V = length × width^2^/2. Mice were euthanized on day 27 post-implantation. The subcutaneously transplanted tumors were dissected, weighed, recorded, and photographed. The animal experiments were approved by the Xiangya Medical School’s Institutional Animal Care and Use Committee.

### 2.12. Flow Cytometry Analysis

#### 2.12.1. Apoptosis

Sh-FTSJ1 or control shRNA were transfected into MDA-MB-231 and HCC1937 cells with PEI (Sigma, Osterode am Harz, Germany). After 6 h, the culture medium was renewed. After 48 h, carefully transfer the supernatant of the flow-through to a new microcentrifuge tube and adherent cell digested with EDTA-free pancreatic enzymes for 3–5 min at 37 °C. Supernatant and digested cell suspension were mixed together and centrifuged at 1000 rpm for 3 min, then washed twice with pre-cooled PBS, and then resuspended with 1× binding buffer, and incubated with Annexin V-FITC and PI staining (Vazyme, A213-01, Vazyme International LLC, Nanjing, China) at 20~25 °C for 10 min. Then add 1× binding buffer and mix gently. The samples were then analyzed by flow cytometry.

#### 2.12.2. Co-Culture Assay

After 24 h of co-culture, the T cells in the supernatant were collected for flow antibody staining as follows: FTTC anti-human CD3 antibody (Cat: 317306, Biolegend, San Diego, CA, USA), PE/Cyanine7 anti-human CD8 antibody (Cat: 344712, Biolegend, USA), Brilliant Violet 421 anti-human granzyme B antibody (Cat: 563389, BD biosciences, Franklin Lakes, NJ, USA), Alexa Fluor 647 anti-human perforin (Cat: 563576, BD Biosciences, USA), Zombie Aqua Fixable Viability Kit (Cat: 423101, Biolegend, USA).

#### 2.12.3. Xenografts Tumors Infiltrating Lymphocyte Analysis

Xenografts tumors were mechanically disrupted and digested into single-cell suspension by collagenase IV (Cat: A004186, Sangon Biotech, Shanghai, China) and DNase I (Cat: EN0521, Thermo Fisher Scientific, USA). A cell strainer with a pore size of 70 μm (Biosharp, Hefei, China) was utilized to filter out impurities. T cells in the supernatant were collected for flow antibody staining as follows: PE anti-mouse CD45 antibody (Cat: 147712, Biolegend, USA), PE/Cyanine7 anti-mouse CD3 antibody (Cat: 100220, Biolegend, USA), FITC anti-mouse CD8a antibody (Cat: 100706, Biolegend, USA), PerCP/Cyanine5.5 anti-mouse CD4 antibody (Cat. No. 116012, Biolegend, USA), Zombie Aqua Fixable Viability Kit (Cat: 423101, Biolegend, USA).

### 2.13. Clinical Specimens

Our study, approved by the Research Ethics Committee of Xiangya Hospital, incorporated a total of 120 TNBC tissue samples. We gathered encoded clinicopathological data and follow-up information from newly diagnosed TNBC patients who received standard treatment at Xiangya Hospital, Central South University, during the period from May 2012 to October 2017.

### 2.14. Immunohistochemical and Immunofluorescence Staining

We performed immunohistochemical (IHC) staining to evaluate the expression of FTSJ1 in TNBC cancer samples. The TNBC tissue sections, with a thickness of 4 μm, underwent deparaffinization and rehydration. Antigen retrieval was conducted using Tris-EDTA buffer at pH 9.0. Following this, the sections were initially treated with a 3% H_2_O_2_ solution for 10 min and then blocked with 5% goat serum for 1 h. We then applied primary FTSJ1 antibodies (Abcam, ab227259) to the slides, allowing for overnight incubation at 4 °C [24]. On the subsequent day, we treated the slides using Two-Step IHC reagents and applied the 3,3-diaminobenzidine (DAB) solution in accordance with the manufacturer’s instructions. Counterstaining was performed using Harris modified hematoxylin. We classified the staining intensity of the samples into four categories: negative (0), weak (1), moderate (2), and strong (3). Simultaneously, the assessment of the percentage of positive cells was conducted using the following scale: 0 (0–5%), 1 (5–25%), 2 (26–50%), 3 (51–75%), or 4 (>75%).

We determined the total score for each sample by multiplying the staining intensity and positive cell scores. Subsequently, the samples were categorized into high FTSJ1 and low FTSJ1 groups based on the expression of the FTSJ1 protein. In the immunofluorescence (IF) assay, a multi-color IF of the tissue sample was conducted using a multiple fluorescent IHC staining kit (Absin, China). Anti-CD8 (Cat: ab4055, Abcam, USA) and anti-perforin (Cat: ab268108, Abcam, USA) antibodies were utilized in this process.

### 2.15. Statistical Analysis

We conducted all in vitro experiments in triplicate, ensuring a minimum of three repetitions. The data were expressed as the mean ± the standard error of the mean (SEM). Group differences were analyzed using either a *t*-test or analyses of variance (ANOVAs). KM analysis was employed to generate survival curves, with the significance determined by the log-rank test. A *p*-value of less than 0.05 was deemed statistically significant. Graphing and statistical analyses were performed using Prism 8.4.2 (GraphPad Software, San Diego, CA, USA).

## 3. Results

### 3.1. FTSJ1 Is Associated with Poor Prognosis in TNBC

To define the role of FTSJ1 in cancer, we analyzed the RNA-seq data collected from the TCGA database, and we found that the expression of FTSJ1 was significantly upregulated in BRCA as compared with their adjacent normal tissues (Figure 1A). For major subclasses of BC, relatively higher FTSJ1 expression was observed in TNBC and HER2-positive cells than in luminal BRCA cells (Figure 1B).

What is more, GSVA analysis based on TCGA-BRCA dates shows high FTSJ1 expression closely related to the activation of signaling pathways such as MYC, epithelial–mesenchymal transition (EMT), TGF-β, DNA repair, IL6-JAK-STAT3, PI3K/AKT/mTOR, and Notch that promotes tumor occurrence and development, which suggests that FTSJ1 may play a certain role in the occurrence and development of TNBC (Figure 1C).

To further detect the correlations of FTSJ1 expression with prognosis, OS KM plots, were generated using the GEPIA2 “Survival Analysis” module. Patients were divided into two groups based on the median expression level of FTSJ1. OS analysis demonstrated that the patients with BRCA in the low FTSJ1 expression group had significantly longer OS than those in the high FTSJ1 expression group (*p* < 0.05) (Figure 1D). We speculate that FTSJ1 might play a crucial role in BRCA initiation and progression.

We then examined FTSJ1 protein levels in 120 human primary breast tumor specimens by IHC staining. A high FTSJ1 expression based on IHC was correlated with reduced survival in TNBC patients (*p* = 0.0343) (Figure 1E). Representative IHC is shown in Figure 1F. Subgroup comparisons for overall survival of the prespecified subgroup’s T stage, N stage, age, Ki67, and expression of FTSJ1 are shown in Figure 1G in the form of a forest plot. High FTSJ1 can significantly increase the occurrence of poor outcomes compared to low FTSJ1, indicating it is a risk factor.

### 3.2. FTSJ1 Knockdown Suppresses TNBC Cell Proliferation, Migration and Promotes Apoptosis

In order to investigate the impact of FTSJ1 on the proliferation of TNBC cells, we performed FTSJ1 knockdown (KD) in HCC1937 and MDA-MB-231 cells and established two monoclonal cell lines with stable FTSJ1 knockdown (KD1 and KD2) (Figure 2A). Subsequently, we assessed cell proliferation rates using CCK-8 assays. Our findings revealed that the knockdown of FTSJ1 led to significant suppression of proliferation in both HCC1937 (*p* < 0.0001) and MDA-MB-231 (*p* < 0.0001) (Figure 2B,C). We assessed the impact of FTSJ1 on the migration and metastatic potential of TNBC cells using wound healing assays. The outcomes demonstrated that FTSJ1 knockdown resulted in the inhibition of migration in both HCC1937 (*p* < 0.05) and MDA-MB-231 (*p* < 0.001) (Figure 2D,E). Additionally, flow cytometry analysis revealed that the reduced expression of FTSJ1 significantly increased the apoptosis rate in HCC1937 (*p* < 0.01) and MDA-MB-231 (*p* < 0.01) (Figure 2F,G).

In order to evaluate the impact of FTSJ1 on the in vivo growth of TNBC, we employed short hairpin RNA (shRNA) to knock down the Ftsj1 gene in the mouse-derived TNBC cell line 4T1 (Figure 2H). Subsequently, we inoculated these modified cells into immune-deficient NOD-SCID mice. The knockdown of FTSJ1 resulted in a significant reduction in both tumor volume (*p* < 0.0001) (Figure 2I,J) and weight (*p* < 0.0001) (Figure 2K) compared to tumors derived from unmodified 4T1 cells.

### 3.3. FTSJ1 Knockdown Enhances Tumor-Infiltrating CD8+ T Cell

Based on the TCGA-TNBC database, 24 immune cells were identified using the single-sample gene set enrichment analysis (ssGSEA) algorithm and systematically correlated with FTSJ1 proteins. Among them, we focused on the obvious negative relationship between FTSJ and CD8+ T cells as shown in Figure 3A. Next, we calculated the BRCA CD8+ T-cell-inflamed score and found that the FTSJ1 was significantly negatively correlated (*p* < 0.0001; Figure 3B).

To investigate the relationship between FTSJ1 and tumor-infiltrating CD8+ T cells in vivo, we inoculated control and Ftsj1 KD 4T1 cells into immunocompetent Bal/bc mice. Knockdown of Ftsj1 dramatically decreased the volume (*p* < 0.0001) (Figure 3C,D) and weight (*p* < 0.0001) (Figure 3E) of tumors derived from 4T1 cells. This difference appears to be more pronounced than in immuno-deficient mice, which suggests that the immune system plays a role in the effect of FTSJ1 knockdown on the development of 4T1 xenograft tumors.

To determine the effect of FTSJ1 on CD8+T cell infiltration in TNBC, the allograft tumor was digested into a single-cell suspension and TILs were analyzed using flow cytometry. We found that the proportion of CD8+/CD3+ T cells in the tumor tissue increased from 14.5% to 22.3% after Ftsj1 knockdown (Figure 3F,G) *(p* < 0.05). In addition, the proportions of CD4+T cells showed no significant changes in the tumor tissue in the Ftsj1 KD group compared with the control group (Figure 3F,H).

In the preceding findings, we showcased the immunosuppressive influence of FTSJ1 in TNBC through analyses of external databases and in vitro experiments using mouse TNBC cells. Nevertheless, the association between tumor cells exhibiting high FTSJ1 expression and CD8+ T cells, along with the molecules participating in T cell differentiation, remains unclear at the human tissue level. Hence, we conducted multicolor staining on tumor cells and CD8+ T cells (Figure 4A,B). Additionally, through multi-IF staining, we investigated the relationship between FTSJ1 expression and CD8+ T cell infiltration in clinical TNBC tissues. Our findings revealed a significant negative correlation between FTSJ1 expression and CD8+ T cell infiltration in the TME of TNBC (*p* < 0.0001) (Figure 4C). These results imply that elevated FTSJ1 expression in TNBC may promote the infiltration or presentation of CD8+ T cell clusters with anticancer functionality.

### 3.4. FTSJ1 Knockdown Increases the Sensitivity of TNBC to T Cell Killing

The above suggests that TNBC intrinsic FTSJ1 promotes tumor progression while also attenuating CD8+ T cell infiltration, but its effect on the secretion of its cytotoxic factors remains unclear. Therefore, we established an in vitro co-culture model to simulate the process of T-cell-killing target cells. FTSJ1 KD/OE, control KD/OE MDA-MB-231-GFP cells were co-cultured with preactivated human T cells (Figure 5A) with three E:T ratios (0:1, 10:1, and 20:1) for 12–48 h. Adherent MDA-MB-231-GFP cells were observed with a fluorescence microscope. The representative view is shown in Figure 5B. The knockdown and over expression efficiency of FTSJ1 protein in MDA-MB-231-GFP are shown in Figure 5C. We found that the killing rate (one count of surviving cells in the co-cultured group/count of cells in the untreated group)% of the FTSJ1 KD group was higher than that of the control group under two different ET ratios (*p* = 0.0008, *p* < 0.0001) (Figure 5D). On the contrary, the overexpression of FTSJ1 makes it harder for the TNBC cells to be killed by T cells (*p* = 0.0051, *p* < 0.0001) (Figure 5E). Evidently, FTSJ1 knockdown increases the sensitivity of TNBC to T cell killing. Additionally, the outcome of the T-cell-mediated cancer cell killing assay (without T cells group, E:T ratios = 0:1) can infer that FTSJ1 plays an oncogenic role in TNBC.

To assess CD8+T function in co-culture, we collected suspended T cells from the co-cultured supernatant and analyzed the ability to secrete cytokines using flow cytometry (Figure 5A,H). Although there was no significant difference in granzyme B (GZMB) secretion of CD8+ T cells in the KD or OE group, OE of FTSJ1 on tumor cells directly suppressed perforin secretion function of T cells (*p* = 0.0047) while KD FTSJ1 on tumor cells significantly reversed the tendency (*p* = 0.0149) (Figure 5F–H).

In addition, we also verified the correlation of FTSJ1 with perforin+ CD8+ T cells in clinical TNBC tissues (Figure 4A,B). We found that high FTSJ1 expression had a significant negative correlation with perforin+ CD8+ T cell infiltration in the TME of TNBC (*p* < 0.002) (Figure 4D).

So, we can draw the conclusion that FTSJ1 knockdown increases the sensitivity of TNBC to T cell killing and regulates the secretion of perforin in T cells.

Tumor cells express immune checkpoints such as the ligand of programmed death protein 1 (PD-L1), cytotoxic T lymphocyte-associated antigen-4, (CTLA-4), and lymphocyte activation gene 3 (LAG3), which inhibit the immune response. Additionally, immune-regulatory molecules released by tumor cells, such as indoleamine 2,3-dioxygenase 1 (IDO1), can impede effective T cell function. In order to understand if FTSJ1 exerts its immunosuppressive effects by modulating the aforementioned immune inhibitory molecules. We analyzed the association between FTSJ1 and these molecules in BRCA. Finally, we found that FTSJ1 expression was positively correlated with most immunosuppressive molecules in BRCA (Figure 5I). Among them, the most closely related are CD274, TNSF4, TNSF9, and CD70. Consequently, in tumor cells after co-culture, we found that mRNA expression of CD274 was significantly downregulated in FTSJ1 KD cell lines (Figure 5J) and upregulated in FTSJ1 OE cell lines compared to the control cell line (Figure 5K). Unexpectedly, we also found that PD-L1 is significantly upregulated in FTSJ1 OE cell lines despite no significant downregulation in FTSJ1 KD cell lines. This suggests that CD276 may be regulated by FTSJ1 and PD-L1 may be involved in the immunosuppressive effects of upregulation of FTSJ1. This also suggests that the immunosuppressive effect of FTSJ1 may be the result of the superposition of multiple factors regulated by FTSJ1. The heatmap (Figure 5L) displays the relative Log_2_ fold changes (Log_2_ FC) in genes in the two groups.

## 4. Discussion

FTSJ1 is a member of the methyltransferase superfamily and is involved in ribosomal RNA processing and modification. FTSJ1 plays an important role in brain development and mutation of FTSJ1 is related to intellectual disability [12,13,14,15]. The role of FTSJ1 in cancer is elusive. In this study, we comprehensively analyzed the role of FTSJ1 in TNBC.

In the present study, our analysis of public databases showed that the expression of FTSJ1 is higher in BRCA than in paired normal tissue, and high expression of FTSJ1 is associated with poorer OS. We also found high expression of FTSJ1 closely associated with many signaling pathways that promote tumorigenesis and development. In addition, we validated that FTSJ1 is associated with poor outcomes using TNBC tissue samples. Knockdown of FTSJ1 suppressed proliferation and migration but directly induced increased apoptosis of MDA-MB-231 and HCC1937. The effect of promoting tumor growth is also reflected in mouse models of TNBC. In short, our study demonstrates for the first time that FTSJ1 is oncogenic in TNBC.

In initial investigations, TILs found in breast cancer were characterized as predominantly consisting of cytotoxic (CD8+) T cells. They also include differing amounts of helper (CD4+) T cells, CD19+ B cells, and infrequent natural killer (NK) cells within their composition [25,26]. Multiple findings suggest that a higher presence of TILs within the tumor’s supportive tissue is linked to an increased likelihood of successful recovery in individuals diagnosed with early-stage TNBC- and HER2-positive breast cancer [27]. Augmented TILs correspond to a greater chance of achieving pathological complete response (pCR) and enhanced survival rates when treated with anthracycline-based chemotherapy and trastuzumab in patients specifically categorized under these disease subtypes (TNBC and HER2-positive, respectively) [27,28]. Furthermore, the correlation between elevated levels of tumor-infiltrating lymphocytes (TILs) and a more favorable outlook for individuals diagnosed with early-stage breast cancer has been validated across a substantial cohort comprising thousands of patients [29,30,31,32,33]. In the present study, we demonstrate that knockdown of FTSJ1 is associated with higher CD8+ T cell infiltration in mouse models of TNBC. Also, the relationship between FTSJ1 and CD8+ T cell infiltration was verified using clinical tissue specimens. Apart from the above, a direct in vitro co-culture model simulating the T cell killing process was used to study the effect of FTSJ1 on the cytokine secretion capacity of CD8+ T cells in our study. We discovered that knocking down FTSJ1 in TNBC cells increases the sensitivity of TNBC cells to T-cell-mediated cytotoxicity and significantly suppresses the secretion of perforin by T cells, which co-cultured with them. Similarly, IF experiments on clinical specimens verified this. Taken together, our study suggests that FTSJ1 plays an immunosuppressive role in the TME of TNBC.

TNBC, known for its aggressive nature and significant intra-tumoral heterogeneity, often exhibits resistance to various therapies [34]. Immunotherapy is quickly becoming a conventional and widely accepted treatment method for solid tumors [35,36]. However, this approach proves beneficial for fewer than 30% of patients [37]. In clinical trials, it was observed that the combination of anti-PD-1 treatment and albumin-bound paclitaxel yielded clinically significant overall survival benefits exclusively in patients with PD-L1-positive tumors [38].

There is still a lot to discover regarding the mechanisms of action of immunotherapy and the optimal selection of treatment or combinations for individual patients. Grasping the mechanisms and molecular foundations of effective anti-tumor immune responses is crucial for advancing novel immunotherapeutic strategies. Tumor-specific CD8+ T cells possess the capability to identify and eliminate cancer cells, yet they frequently face functional limitations within the TME [39,40]. The impaired functionality of CD8+ T cells poses not just a challenge but also a hindrance to the development of effective anti-tumor immunity [41]. In our research, elevated FTSJ1 expression hampers T cell function by suppressing perforin activity.

CD276 (B7-H3), belonging to the B7 family, was initially described by Chapoval et al. in 2001 [42]. The B7 family, crucial in modulating T cell immune responses, has garnered attention due to its potential therapeutic implications in cancer treatment. B7 family members, encompassing B7-H1 (PD-L1/CD274), B7-DC (PD-L2/CD273), B7-H6 (NCR3LG1), B7-H7 (HHLA2), B7.1 (CD80), and B7.2 (CD86), can be categorized based on their influence on T cell activation as either (a) providing co-stimulation, (b) exerting co-inhibitory effects, or (c) having a combination of both co-stimulatory and co-inhibitory properties [43]. Newly conducted research has identified CD276 acts as a novel immune checkpoint molecule, a suppressor of T cells, and is directly involved in modulating the immune response and various malignant biological behaviors exhibited by tumor cells. Higher expression levels of CD276 have been found to be correlated with poorer prognosis in BRCA patients [44,45]. Our study suggests that FTSJ1 may regulate the expression of CD276, which partly explains the effect of FTSJ1 on tumor and tumor immunity. However, the causes of malignant behavior and the immunosuppression of tumors are complex, and intercellular communication is constantly changing, so the final outcome may not be influenced by a single gene or pathway, which needs further exploration.

## 5. Conclusions

In summary, our study demonstrates that the knockdown of FTSJ1 in TNBC cells not only suppresses the proliferation and migration and induces apoptosis of cancer cells but also increases the sensitivity of TNBC cells to T-cell-mediated cytotoxicity. The high expression attenuates CD8+ T cell infiltration and is related to TNBC patients’ poor prognosis. FTSJ1 acts as a tumor promotor, is involved in cancer immune evasion, and may serve as a potential immunotherapy target in TNBC.

## Figures and Tables

**Figure 1 cancers-16-00597-f001:**
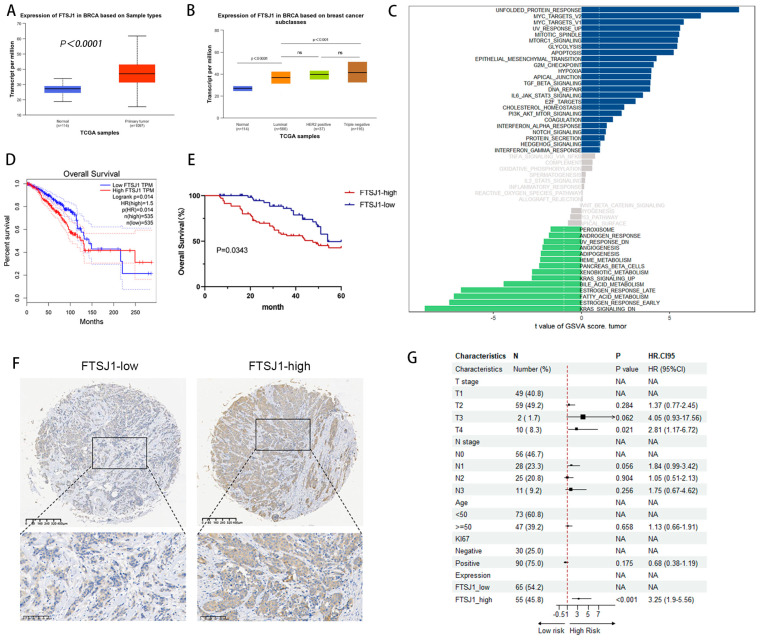
FTSJ1 is significantly overexpressed in aggressive TNBC and associated with worse prognosis. (**A**) Expression of FTSJ1 transcript in BRCA compared with their adjacent normal tissues. (**B**) Expression of FTSJ1 transcript in BRCA compared among BRCA subclasses. (**C**) GSVA analysis based on TCGA-BRCA in different signaling pathways. (**D**) KM curves of OS prediction were plotted in BRCA using the GEPIA2 “Survival Analysis” module. (**E**) Overall survival of TNBC patients with different FTSJ1 protein expressions. Log-rank test was utilized to statistically evaluate the disparities in survival curves. (**F**) Representative images of IHC staining for FTSJ1 protein expression in paraffin-embedded human TNBC tissues. Scale bars, 400 μm (low magnification view), 200 μm (high magnification view). (**G**) Subgroups effect on overall survival by forest plot. HR = hazard ratio.

**Figure 2 cancers-16-00597-f002:**
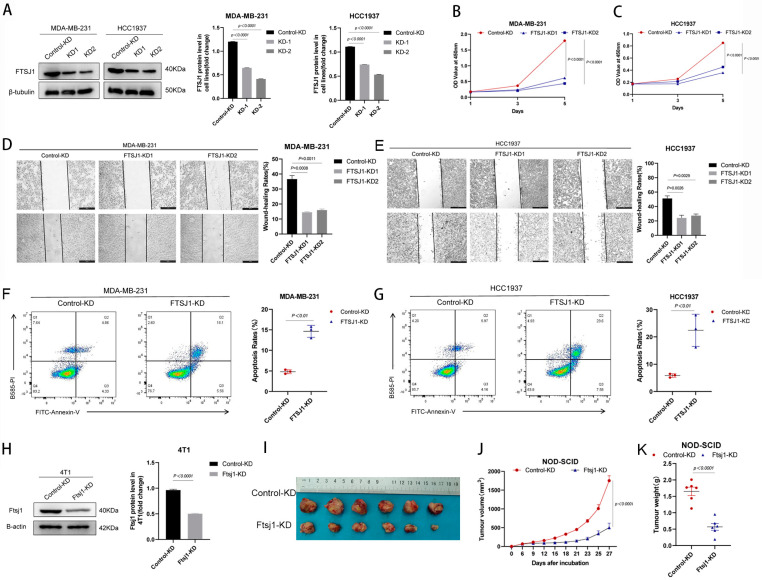
FTSJ1 knockdown suppresses TNBC cell proliferation and migration and promotes apoptosis. (**A**) Knockdown efficiency of FTSJ1 protein in HCC1937 and MDA-MB-231. The Image Lab 6.0.1 software (Bio-Rad, Hercules, CA, USA) was employed to quantitate and analyze blots’ grayscale values. The uncropped bolts are shown in Appendix A. (**B**,**C**) Proliferation rate of FTSJ1 KD and control TNBC cells. (**D**,**E**) Scratch wound healing assay in FTSJ1 KD and control HCC1937 or MDA-MB-231 cells were evaluated. Statistical analysis was performed using unpaired *t*-test. (**F**,**G**) Flow cytometry analysis for FTSJ1 knockdown-induced apoptosis in HCC1937 and MDA-MB-231 cells using an unpaired *t*-test. (**H**) Knockdown efficiency of Ftsj1 protein in 4T1. FTSJ1 knockdown inhibits the growth of 4T1 cells in NOD-SCID mice, the uncropped bolts are shown in Appendix A. (**I**) tumor images, (**J**) growth curve, and (**K**) tumor weight. Statistical analysis in (**H**,**I**) was performed using an unpaired *t*-test. All in vitro experiments were performed in triplicate and repeated at least three times; the results are displayed as mean ± SEM.

**Figure 3 cancers-16-00597-f003:**
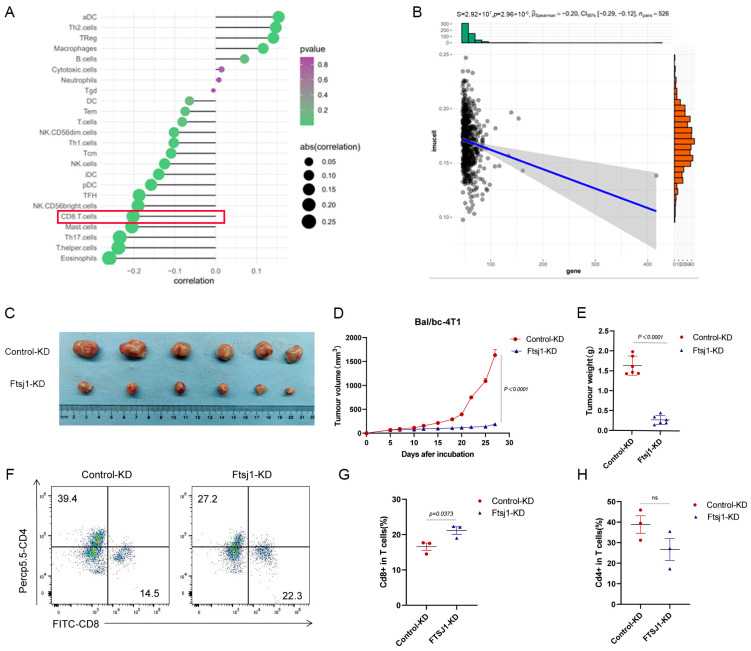
The mouse xenograft model demonstrates that FTSJ1 promotes CD8+ T cell infiltration. (**A**) Correlation between FTSJ1 and tumor-infiltrating 24 immune cells (TIICs) using ssGSEA algorithm in TCGA-BRCA cohort. (**B**) FTSJ1 expression and CD8+ T cell scores in TCGA-BRCA. FTSJ1 knockdown inhibits the growth of 4T1 cells in Balb/c mice (**C**) tumor images, (**D**) growth curve, and € tumor weight. (**F**–**H**) Flow cytometry analysis of cd8+ and cd4+ T cells in tumor infiltration in the FTSJ1 knockdown group versus controls. (**D**,**E**,**G**,**H**) was performed using a two-sided unpaired *t*-test. All in vitro experiments were performed in triplicate and repeated at least three times, the results are displayed as mean ± SEM.

**Figure 4 cancers-16-00597-f004:**
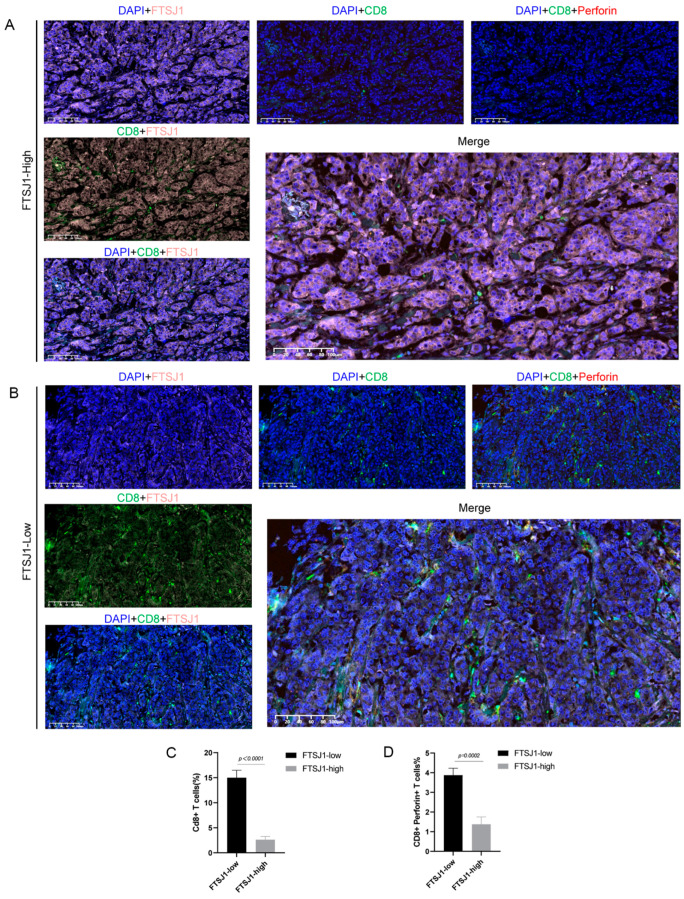
Tissue specimens demonstrate high FTSJ1 expression negatively correlated with CD8+ T cell infiltration. Representative images of immunofluorescence multi-staining for both FTSJ1, CD8, and perforin in TNBC tissues. Scale bars, 100 μm. (**A**) FTSJ1-high group, (**B**) FTSJ1-low group. (**C**,**D**) Statistical analysis was conducted by using an unpaired *t*-test. Results are presented as mean ± SEM.

**Figure 5 cancers-16-00597-f005:**
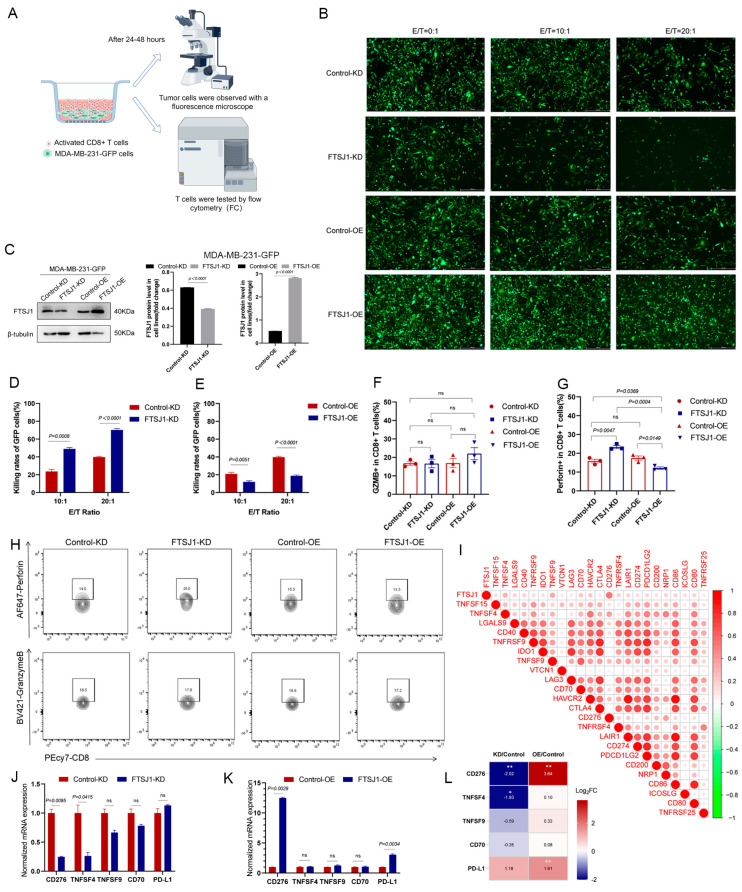
FTSJ1 knockdown increases the sensitivity of TNBC to T cell killing. (**A**) In vitro co-culture analysis model diagram by Figdraw. (**B**) Representative field of view of fluorescent tumor cells after co-culture. Scale bars, 450 μm. (**C**) Knockdown and overexpression efficiency of FTSJ1 protein in MDA-MB-231-GFP. The Image Lab 6.0.1 software (Bio-Rad, CA, USA) was employed to quantitate and analyze blots’ grayscale values. The uncropped bolts are shown in Appendix A. (**D**,**E**) The comparison of cytotoxicity rates between the KD/OE group and the control group. Statistical analysis was conducted by using a two-sided unpaired *t*-test. Results are presented as mean ± SEM. (**F**,**G**) Flow cytometry statistical analysis was conducted by using a chi-square (math.) test. (**H**) Flow cytometry analysis of Perforin+cd8+ and GZMB+cd8+ T cells after co-culture. (**I**) Correlation between FTSJ1 T cell-inflamed and immune checkpoint inhibitor genes. Positive correlation was marked in red, while negative correlation was marked in green. (**J**,**K**) The normalized mRNA expression levels of CD276, TNSF4, TNSF9, CD70, and PD-L1 in control KD, FTSJ1 KD, control OE, and FTSJ1 OE MDA-MB-231-GFP cells. (**L**) The heatmap displays the relative log_2_ fold changes in genes in the two groups. All in vitro experiments were performed in triplicate and repeated at least three times; data are presented as mean ± SEM. Significance levels are indicated as * *p* < 0.05, ** *p* < 0.01. Original western blots are presented in Appendix A.

## Data Availability

The data can be shared up on request.

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
