# Peer review of "Triple-Negative Breast Cancer Intrinsic FTSJ1 Favors Tumor Progression and Attenuates CD8+ T Cell Infiltration"

_cancers, 2024, doi:10.3390/cancers16030597_

Round 1

Reviewer 1 Report

Comments and Suggestions for Authors

Figure 1 frames D and E both address overall survival. However, frame D has the high vs low expression lines overlap, whereas frame E lacks overlap. This is confusing and needs clarification.

The in vivo studies reported involve murine mammary tumors. It is unclear how relevant this murine tumors are to human breast cancer. Human PDX would be more convincing.

Author Response

Thank you very much for taking the time to review this manuscript. Please find the detailed responses below and the corresponding revisions s in the re-submitted files.

Comments 1: Figure 1 frames D and E both address overall survival. However, frame D has the high vs low expression lines overlap, whereas frame E lacks overlap. This is confusing and needs clarification.

Response 1: Thank you for your nice comments. Fistly, Figure 1D were generated using the GEPIA2 “Survival Analysis” module. OS analysis demonstrated that the patients with BRCA which includes all types of breast cancer. This analysis provides insight into the prognostic impact of FTSJ1 in patients with all types of breast cancer. However, the overall survival data presented in Figure 1E are based on patients with triple-negative breast cancer. Second, the follow-up time in Figure 1D reaches 300 months, and the overlap occurs after 200 months of follow-up. The follow-up time in Figure 1E is 60 months and is broadly consistent with the trend in Figure 1D for this follow-up period.

Comments 2: The in vivo studies reported involve murine mammary tumors. It is unclear how relevant this murine tumors are to human breast cancer. Human PDX would be more convincing.

Response 2: We gratefully appreciate for your valuable suggestion. It is really true as reviewer suggested that human PDX would be more convincing. But for this article, the use of mouse-derived tumor cells in this paper is based on the following reasons: Fistly, 4T1 is a mammary tumor cell line that originates from spontaneous mammary tumors in BALB/c mice. These highly aggressive and tumorigenic cells behave very similarly to human breast cancer, in terms of growth and metastatic spread. Specifically, triple-negative breast cancer (TNBC) was investigated in the 4T1 tumor model. Second, we want to understand whether the intact immune system plays a role in the effect of FTSJ1 on tumor progression, so the use of mouse-derived 4T1 cells is necessary to mimic the immune microenvironment in immunocompetent mice. Third, to understand the role of immunity, we must use conspecific cells for comparison in immunocompetent(Figure 3C) and immunodeficient(Figure 2I) mouse models.

Reviewer 2 Report

Comments and Suggestions for Authors

Dear authors,

Congratulations for your hard work!

You demonstrate that FTSJ1 serves as a tumor promotor in triple- negative breast cancer and knockdown of FTSJ1 inhibits triple-negative breast cancer proliferation, migration, induces apoptosis and increase the sensitivity of TNBC cells to T cell–mediated cytotoxicity. What makes it more interesting is that FTSJ1 expression has been associated with CD8+T cell infiltration in the tumor microenvironment in triple-negative breast cancer and also with patients’ poor prognosis.

However, I have Some questions/comments:

 I. In figure 2A, the presentation of the results is confusing and difficult to follow.

1.       In the experiments presented in this figure, the authors probably report the result of two KD experiments of the FTSJ1-KD protein (KD1 and KD2) for each cell line. If so, why they used only one control? The authors should explain how the experiments were done, what KD1 and KD2 mean and how many experiments were done. To be easier to understand, in the first picture of figure 2A, MDA-MB-231-control and HCC1937-control should be replaced with Control-KD, as used in the other graphs presented in figure 2A.

2.       To examine the knockdown efficiency of FTSJ1 protein in HCC1937 and MDA-MB-231 cells and in HCC1937 cells, B-Actin is used as a loading control in western blotting experiments (Figure 1A, first picture). The reason for commonly chosen B-Actin protein as a loading control is the protein generally expression across all eukaryotic cells. Another reason for using this protein as a loading control, is the fact that its expression levels do not vary due to cellular treatment. In this study, knockdown of FTSJ1 induced an increase of B-Actin expression in both cell lines. How can this be explained and how this increase may be related to the FTSJ1 function in the studied cells. To ensure the accuracy of the results, another housekeeping proteins ( e.g. β-tubulin, Lamin B1)  should be used as loading controls as internal reference.

3.       The FTSJ1 quantitate was probably assessed with the Image Lab 6.0.1 software, that should be mentioned in the figure legend of the figure 2.

 II. In figure 5C, the level of FTSJ1 expression is much lower in In Control-OE than in control-KD, although the expression of GAPDH is higher in Control-OE than in control-KD, which is difficult to understand. The authors should have an explanation for these experiments.

 III. In the figure legends, it should also be mentioned the number of experiments used to calculate the SEM.

 IV. There is no rule in the use of abbreviations, for some abbreviations the spelled-out version of abbreviation is presented in the abstract, for others in the introduction. Some abbreviations are accompanied by the spelled-out version several times, e.g. tumor microenvironment (TME), while others are not explained at all (e.g. LIN9; IF1α , Elp3, , Control-OE, FTSJ1-OE).

Normally, the first time an abbreviation is used in the text, it is present both the spelled-out version and the abbreviation form and after that, only the abbreviation is used.

 Best regards

Author Response

Thank you very much for taking the time to review this manuscript. Please find the detailed responses below and the corresponding revisios in the re-submitted files.

Comments 1-(1): In the experiments presented in this figure, the authors probably report the result of two KD experiments of the FTSJ1-KD protein (KD1 and KD2) for each cell line. If so, why they used only one control? The authors should explain how the experiments were done, what KD1 and KD2 mean and how many experiments were done. To be easier to understand, in the first picture of figure 2A, MDA-MB-231-control and HCC1937-control should be replaced with Control-KD, as used in the other graphs presented in figure 2A.

Response 1-(1): Thank you for pointing this out. We are very sorry for the omission in my explanation of KD1 and KD2, which are two monoclonal cell lines with stable FTSJ1 knockdown in both two TNBC cell lines, so the same knockdown control was used. This has been illustrated in the revised manuscript (line 323-324). All in vitro experiments were performed in triplicate and repeated at least three times, which has been explained in the method section (line 275-276). The identification of the control group in Figure 2 A has been revised in the new version as suggested.

Comments 2: To examine the knockdown efficiency of FTSJ1 protein in HCC1937 and MDA-MB-231 cells and in HCC1937 cells, B-Actin is used as a loading control in western blotting experiments (Figure 1A, first picture). The reason for commonly chosen B-Actin protein as a loading control is the protein generally expression across all eukaryotic cells. Another reason for using this protein as a loading control, is the fact that its expression levels do not vary due to cellular treatment. In this study, knockdown of FTSJ1 induced an increase of B-Actin expression in both cell lines. How can this be explained and how this increase may be related to the FTSJ1 function in the studied cells. To ensure the accuracy of the results, another housekeeping proteins ( e.g. β-tubulin, Lamin B1)  should be used as loading controls as internal reference.

Response 2: Thank you again for your positive comments and valuable suggestions to improve the quality of our manuscript. We are very sorry for our negligence of changes in the expression of the β-Actin protein and focused only on the changes in FTSJ1. According to your suggestions, we repeated the experiment and used β-tubulin as loading controls. We have supplemented new data in Figure 1A and Figure S1.

Comments 3-(1): The FTSJ1 quantitate was probably assessed with the Image Lab 6.0.1 software, that should be mentioned in the figure legend of the figure 2.

Response 3-(1): Thank you for your careful reminder on our article. According to your suggestions, we have add this in the figure legend of the figure 2 and figure 5 using red-colored text in our previous draft.

Comments 3-(2): In figure 5C, the level of FTSJ1 expression is much lower in Control-OE than in control-KD, although the expression of GAPDH is higher in Control-OE than in control-KD, which is difficult to understand. The authors should have an explanation for these experiments.

Response 3-(2): Thank you for your nice comments on our article. The level of FTSJ1 expression in control-OE and control-KD were run on the different blot and developed under different exposure conditions. We are very sorry for our negligence that  high-contrast blots (control-OE/FTSJ1-KD) is improper, as overexposure may mask additional bands. We repeated the experiment and the four treatments were subjected to the same conditions of protein extraction, gel running and development, and the loading volume was adjusted. We have supplemented new data in Figure 5C and Figure S1.

Comments 3-(3): In the figure legends, it should also be mentioned the number of experiments used to calculate the SEM.

Response 3-(3):Thank you again for your valuable suggestions to improve the quality of our manuscript. These have been added to the the figure legends of revised manuscript.

Comments 3-(4): There is no rule in the use of abbreviations, for some abbreviations the spelled-out version of abbreviation is presented in the abstract, for others in the introduction. Some abbreviations are accompanied by the spelled-out version several times, e.g. tumor microenvironment (TME), while others are not explained at all (e.g. LIN9; IF1α , Elp3, , Control-OE, FTSJ1-OE).Normally, the first time an abbreviation is used in the text, it is present both the spelled-out version and the abbreviation form and after that, only the abbreviation is used.

Response 3-(4):Thank you for pointing this out. We have made corresponding correction according to you suggestion.

Round 2

Reviewer 2 Report

Comments and Suggestions for Authors

Dear authors,

Congratulations for your hard and interesting work!

Best regards